# DMC-VB: A Benchmark for Representation Learning for Control with Visual Distractors

**Joseph Ortiz**\*, **Antoine Dedieu**\*, **Wolfgang Lehrach, J. Swaroop Guntupalli,**
**Carter Wendelken, Ahmad Humayun, Guangyao Zhou, Sivaramakrishnan Swaminathan,**
**Miguel Lázaro-Gredilla, Kevin Murphy**
Google DeepMind
`{joeortiz,adedieu}@google.com`

## Abstract

Learning from previously collected data via behavioral cloning or offline reinforcement learning (RL) is a powerful recipe for scaling generalist agents by avoiding the need for expensive online learning. Despite strong generalization in some respects, agents are often remarkably brittle to minor visual variations in control-irrelevant factors such as the background or camera viewpoint. In this paper, we present the *DeepMind Control Vision Benchmark* (DMC-VB), a dataset collected in the *DeepMind Control Suite* to evaluate the robustness of offline RL agents for solving continuous control tasks from visual input in the presence of visual distractors. In contrast to prior works, our dataset (a) combines locomotion and navigation tasks of varying difficulties, (b) includes static and dynamic visual variations, (c) considers data generated by policies with different skill levels, (d) systematically returns pairs of state and pixel observation, (e) is an order of magnitude larger, and (f) includes tasks with hidden goals. Accompanying our dataset, we propose three benchmarks to evaluate representation learning methods for pretraining, and carry out experiments on several recently proposed methods. First, we find that pretrained representations do not help policy learning on DMC-VB, and we highlight a large representation gap between policies learned on pixel observations and on states. Second, we demonstrate when expert data is limited, policy learning can benefit from representations pretrained on (a) suboptimal data, and (b) tasks with stochastic hidden goals. Our dataset and benchmark code to train and evaluate agents are available at https://github.com/google-deepmind/dmc_vision_benchmark.

## 1 Introduction

Reinforcement learning (RL) [46] provides a framework for learning behaviors for control, represented by policies, that maximize rewards collected in an environment. Online RL algorithms iteratively take actions—collecting observations and rewards from the environment—then update their policy using the latest experience. This online learning process is however fundamentally slow. Recently it has become clear that learning behaviors from large previously collected datasets, via behavioral cloning or other forms of offline RL [29], is an effective alternative way to build scalable generalist agents—see e.g. [40, 38].

Despite these advances, recent research indicates that agents trained on offline visual data often exhibit poor generalization to novel visual domains, and can fail under minor visual variations in the background or camera viewpoint [9, 41]. This poor generalization can be understood by formalizing environments as Markov Decision Processes (MDPs) with a factorized state consisting of control relevant and irrelevant variables [13]—see Appendix B for further discussion. In such MDPs, in

---

\*Equal contribution

38th Conference on Neural Information Processing Systems (NeurIPS 2024) Track on Datasets and Benchmarks.

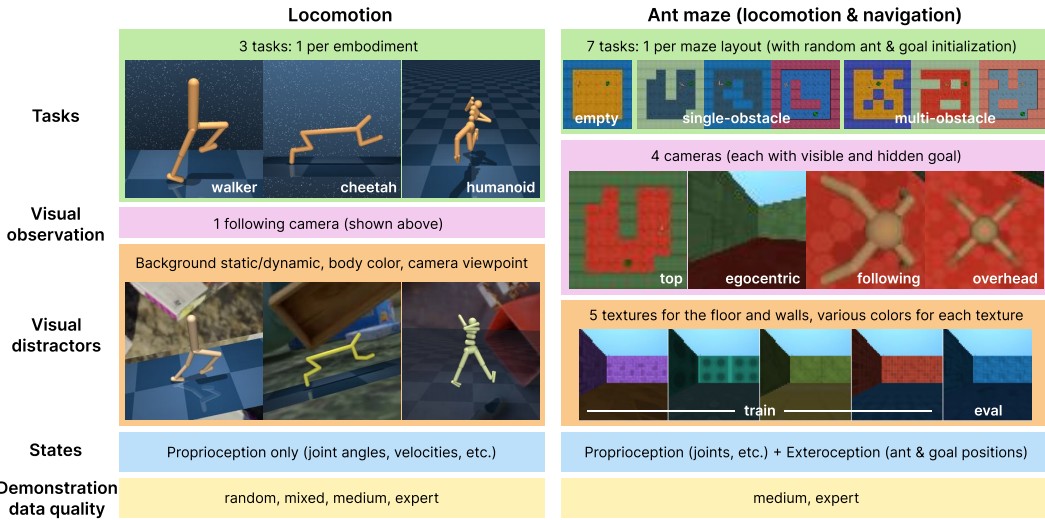

Figure 1: DeepMind Control Vision Benchmark.

order to generalize to novel visual domains, an agent must learn latent representations that capture the information sufficient for control, yet that are minimal and therefore robust to irrelevant variations in the input—see [27, 44]. We call such representations "*control-sufficient*".

Several works derive latent representations for control with generative models of the input [49, 48]. By construction, such representations are not minimal, and may not be robust. Therefore various non-generative methods have been proposed as alternatives. These methods include contrastive learning [11, 28], latent forward models [21, 39], and inverse dynamics models [26, 25, 7]. While there exist several datasets to evaluate the generalization of these (and other) representation learning techniques to various visual perturbations [9, 41, 30, 50], these datasets do not provide the essential properties that are needed to thoroughly evaluate these representation learning methods for control.

To fill this gap, we introduce the *DeepMind Control Vision Benchmark* (DMC-VB)—a dataset collected using the DeepMind Control Suite [47] and various extensions thereof. DMC-VB is carefully designed to enable systematic and rigorous evaluation of representation learning methods for control in the presence of visual variations, by satisfying six key desiderata. (a) It contains a diversity of tasks—including tasks where state-of-the-art algorithms struggle—to drive the development of novel algorithms. (b) It contains different types of visual distractors (e.g. changing background, moving camera) to study robustness to various realistic visual variations. (c) It includes demonstrations of differing quality to investigate whether effective policies can be derived from suboptimal demonstrations. (d) It includes both pixel observations and states, where states are relevant proprioceptive and exteroceptive measurements. Policies trained on states can then provide an upper bound to quantify the "*representation gap*" of policies trained on pixels. (e) It is larger than previous datasets. (f) It includes tasks where the goal cannot be determined from visual observations, for which recent work [7] suggests that pretraining representations is critical. As DMC-VB is the first dataset to satisfy these six desiderata (see Table 1) it is well placed to advance research in representation learning for control.

Accompanying the dataset, we propose three benchmarks that leverage the carefully designed properties of DMC-VB to evaluate representation learning methods for control. **(B1)** evaluates the degradation of policy learning in the presence of visual distractors, and, in doing so, quantifies the representation gap between agents trained on states and on pixel observations. We find that the simple behavior cloning (BC) baseline is the best overall method, and that recently proposed representation learning methods, such as inverse dynamics (ID) [25], do not show benefits on our benchmark. **(B2)** investigates a practical setting with access to a large dataset of mixed quality data and a few expert demonstrations. We find that leveraging mixed data for pretraining a visual representation improves policy learning. Finally, **(B3)** studies representation learning on demonstrations with random hidden goals, as may be the case when an agent collects data in a new environment via goal-free exploration. We find that representations pretrained on this data help few-shot policy learning on new tasks.

Table 1: Among existing offline RL datasets for continuous control tasks, DMC-VB is the only one to satisfy the six desiderata we have identified. We highlight the specific benchmark(s) within Sec.5 that leverage each of these properties. This table includes five relevant datasets and is not an exhaustive list. We exclude related environments which do not provide datasets for offline learning.

| Dataset | Task diversity | Distractor diversity | Different policies | States and obs. | Large | Hidden goal |
|---|---|---|---|---|---|---|
| *D4RL* [15] | ✓ | ✗ | (✓) not for each task | ✗ | ✓ | ✓ |
| *VD4RL* [33] | ✗ | (✓) not released | (✓) not released | ✗ | ✗ | ✗ |
| *RL unplugged* [20] | ✓ | ✗ | ✗ | ✗ | ✓ | ✗ |
| *Libero* [31] | ✓ | ✗ | ✗ | ✓ | ✓ | ✗ |
| *Colosseum* [41] | ✓ | (✗) only for eval. | ✗ | ✓ | ✓ | ✗ |
| **DMC-VB** (ours) | ✓ (**B1**) | ✓ (**B1**) | ✓ (**B1-2**) | ✓ (**B1**) | ✓ (**B1-2-3**) | ✓ (**B3**) |

## 2 Related work

**Environments for control:** Many environments have been developed to study the performance of RL agents. The popular *Arcade Learning Environment (ALE)* [4] proposes an interface to hundreds of Atari 2600 games, and has been used in groundbreaking works in deep RL [36, 22]. The *OpenAI Gym* [8] extends *ALE* to board games, robotics, and continuous control tasks; the *DM Control suite* [47] also proposes a similar suite of environments. Several works [45, 23, 53, 2] extend these control environments to visual tasks with various types of distractors, including agent color, background, and camera pose. However, these environments evaluate the robustness of online RL agents to visual distractors and do not provide pre-collected trajectories for offline learning.

**Offline datasets for control:** To foster research in offline RL, *D4RL* [15] proposed a suite of datasets collected in the *OpenAI Gym* which cover diverse tasks with properties such as sparse rewards, partial observability, multitasking, etc. However, as *D4RL* only gives access to states, *VD4RL* [33] extends *D4RL* to complex visual scenes. *VD4RL* considers three locomotion tasks from the *DM Control Suite*, and collects datasets containing observations with visual distractors for each task by deploying five behavioral policies, ranging from random to expert. However, *VD4RL* does not include states, which are useful to measure the representation gap between policies trained on images and on states. Second, the *VD4RL* datasets released only cover a small fraction of the combinations of embodiments, policies and distractors. Finally, each dataset only contains 100k steps which, as we show in Appendix I, can be too small for RL algorithms. The *RL unplugged* [20] datasets also collect trajectories on different environments from *ALE* and the *DM Control Suite*. Recently, *Libero* proposes several datasets to evaluate lifelong learning for robotics manipulation. Finally, *Colosseum* [41] studies generalization on 20 manipulation tasks, which can be systematically modified across 14 axes of variation. However, *RL unplugged*, *Libero*, and *Colosseum* datasets lack visual distractor diversity in the data and do not include different behavioral policies.

**Representation learning for control:** Representation learning methods map a high-dimensional input (an image) into a compact low-dimensional space. Several studies show that pretrained representations can improve RL policy learning [51, 7]. Early works [49, 48] learn representations with auto-encoders. As these models reconstruct the observations, they fail to discard visual distractors and to generalize under visual distribution shifts [9]. To learn control-sufficient representations, several works have explored non-generative losses. Lamb et al. [26] showed that multi-step inverse dynamics (ID) can, theoretically and empirically, learn a "minimal" world model, which can be used for exploration and navigation. ID has also proven to be beneficial to train offline RL agents in the presence of visual distractors [25], and when the goal is a hidden variable [7]. Another approach, latent forward models (LFD), predict future latent representations from current ones [21]. Contrastive methods maximize the similarity between augmentations of the same data [11, 28, 23, 34] or temporally related images [39, 37]. Other works maximize the mutual information between the latents and control relevant variables such as the reward or actions [6, 14] or use masked auto-encoding [35, 42]. Our experiments (Sec. 5) benchmark simple variants of many of these methods on DMC-VB.

# 3 DeepMind Control Vision Benchmark

In this section, we describe our dataset, DMC-VB, that enables systematic evaluation of representation learning methods for control with visual distractors. A visual overview is presented in Fig.1.

## 3.1 Dataset Characteristics / Desiderata

As highlighted in Table 1, DMC-VB is the first offline RL dataset to satisfy these six desiderata:

**Task diversity:** DMC-VB contains both simpler locomotion tasks and harder 3D navigation tasks. We use the same three locomotion tasks as [33]: `walker-walk`, `cheetah-run` and `humanoid-walk`. The walking humanoid task is significantly harder due to its high degree of freedom action space. For each task, the maximum reward is attained by reaching a target velocity. In addition, we create a suite of seven navigation tasks, corresponding to seven maze layouts with three levels of complexity: `empty`, `single-obstacle` ($\times 3$) and `multi-obstacle` ($\times 3$). For each task, a quadruped ant is randomly placed in the maze and has to reach a randomly placed visible goal (a green sphere). We define a dense reward as the negative normalized shortest path length between the ant and the goal, respecting the maze layout—see Appendix C.3. This ensures the reward falls in the range $[-1, 0]$.

**Distractor diversity:** For locomotion tasks, we leverage the *Distracting Control Suite* [45] to add visual variation to the agent color, camera viewpoint, and background. We use three different distractor settings: *none* has no variations; *static* has an image background, random static camera viewpoint and variable agent color; *dynamic* has a video background, randomly moving camera and variable agent color. For backgrounds, we use synthetic videos or images of multiple objects falling from the Kubric dataset [18]. For the navigation tasks, visual distractors consist of different textures and colors applied to floors and walls of the maze environment.

**Demonstration quality diversity:** To produce datasets with diversity in demonstration quality, DMC-VB generates trajectories using behavioral policies of different skill levels, ranging from random to expert. We select checkpoints for each behavioral policy level as follows: *expert* is the first checkpoint that achieves 95% of the reward achieved at convergence, *medium* is the checkpoint that achieves 50% of the reward at convergence, *mixed* uses eight evenly spaced checkpoints that achieve between 0-50% of the reward at convergence, *random* samples actions randomly from the action space. We use the 95% reward checkpoint as the expert policy because we find that behavior diversity diminishes with more training steps. The reward distributions for our dataset are shown in Fig.2.

**State and visual observations:** Few works compare agents trained on pixel observations versus on states. In contrast, DMC-VB systematically collects both states and images which allows training agents on both to systematically quantify the representation gap.

**Large size:** Each dataset in DMC-VB contains 1M steps, which is equivalent to 2k episodes for the locomotion tasks and more than 2k variable-length episodes for the ant maze tasks. This makes DMC-VB one order of magnitude larger than *VD4RL* [33], the most similar prior dataset. We motivate this choice in Appendix I through experiments which find that limiting expert data significantly harms BC agents, particularly for harder tasks and with visual distractors.

**Hidden goal:** [7] shows the benefits of inverse dynamics pretraining for tasks in which the goal is not determinable from the visual observations. Inspired by this finding, we include a variant of our navigation dataset in which the target sphere is not visible, which we study in Sec. 5.3.

## 3.2 Dataset Generation

To generate a dataset of DMC-VB, for each task, we first train an online MPO agent [1] using states for 2M steps. The states and rewards for each task are described in Appendices C.2 and C.3 respectively. Relevant training checkpoints are then selected for the behavioral policies. We generate demonstrations by rolling out the behavioral policy in the environment, collecting rewards, actions, states and visual observations. As in [52, 33], we use an action repeat of two in all environments to increase the change between successive frames and ease learning from pixels.

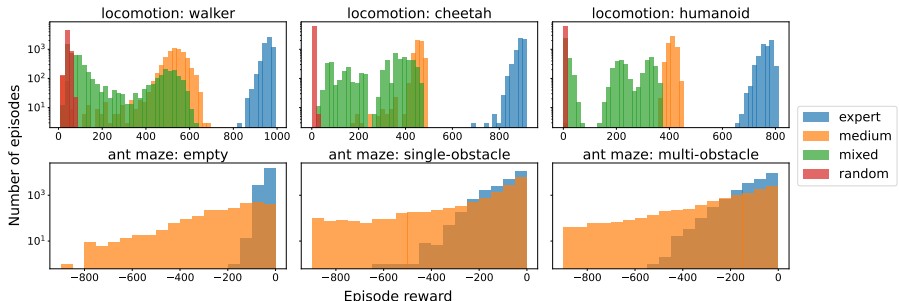

Figure 2: Reward distribution for different behavioral policy levels in DMC-VB. Note the log scale on the vertical axis. Statistics of these distributions are summarized in Appendix C.1.

For navigation tasks, we collect three observations for each timestep: (a) a top-down view of the maze, (b) an egocentric view, (c) a follow-up view[2]—which is useful to infer the agent's state—see Fig.1. We additionally collect these same observations with the goal sphere hidden.

# 4  Agents and Visual Representation Learning for Control on DMC-VB

Given a DMC-VB dataset $D = \{\tau_i\}_{i=1:N}$ of $N$ trajectories pre-collected in an environment, an agent must learn a policy that maximizes the expected sum of the rewards when deployed in the same environment. Each trajectory contains observations, actions and rewards[3]: $\tau = (\mathbf{o}_1, r_1, \mathbf{a}_1, \mathbf{o}_2, r_2, \mathbf{a}_2, ...)$. Given an observation $\mathbf{o}_t$, an agent selects an action $\hat{\mathbf{a}}_t = \pi(\phi(\mathbf{o}_t))$, where $\phi$ is a visual encoder network, and $\pi$ is a policy network.

The agent is trained in a two-stage procedure. The pretraining stage first learns the visual encoder network $\phi$ to minimize a representation learning loss:

$$\phi^* \in \underset{\phi}{\operatorname{argmin}}\ \mathbb{E}_{\tau \sim D}\left[\ \mathcal{L}_{\text{pretrain}}(\phi, \tau)\ \right]\ . \tag{1}$$

The second stage learns the policy network $\pi$ by minimizing a policy learning objective with the visual encoder network $\phi^*$ held constant:

$$\pi^* \in \underset{\pi}{\operatorname{argmin}}\ \mathbb{E}_{\tau \sim D}\left[\ \mathcal{L}_{\text{policy}}(\pi, \phi^*, \tau)\ \right]\ . \tag{2}$$

**Architectures:** We standardize $\phi$ as a convolutional neural network (CNN) followed by a linear projection, and $\pi$ as a multi-layer perceptron (MLP)—see Appendix D for details.

**Frame stacking:** As in [33], we use frame stacking of $\ell = 3$ and select actions as $\hat{\mathbf{a}}_t = \pi(\phi(\tilde{\mathbf{o}}_t))$ where $\tilde{\mathbf{o}}_t = (\mathbf{o}_{t-\ell+1}, \ldots, \mathbf{o}_t)$. We drop the notation $\tilde{\mathbf{o}}_t$ in this section for simplicity. Appendix H verifies, through an ablation study, that frame stacking is crucial for policy learning.

## 4.1  Policy learning

As the primary focus is to benchmark visual representation learning for control, we limit the study to two simple objectives for learning our policy $\pi$: behavioral cloning (BC) and TD3-BC [16].

**Behavioral cloning (BC)** learns a policy via supervised learning by minimizing the objective:

$$\mathcal{L}_{\text{BC}}(\pi, \phi, \tau) = \mathbb{E}_{(\mathbf{o}_t, \mathbf{a}_t) \sim \tau}\left\|\pi(\phi(\mathbf{o}_t)) - \mathbf{a}_t\right\|_2^2\ . \tag{3}$$

**TD3-BC** is a model-free offline RL algorithm that trains an actor and two critic networks [16]. TD3-BC uses the same critic objective as TD3 [17], and adds a BC term to the actor (policy) objective, to regularise the learned policy towards actions in the dataset $D$ (see Appendix D for details):

$$\mathcal{L}_{\text{TD3-BC, Actor}}(\pi, \phi, \tau) = \mathbb{E}_{(\mathbf{o}_t, \mathbf{a}_t) \sim \tau}\left[\lambda\, Q_1(\phi(\mathbf{o}_t), \pi(\phi(\mathbf{o}_t)) - \left\|\pi(\phi(\mathbf{o}_t)) - \mathbf{a}_t\right\|_2^2\right]\ . \tag{4}$$

---

[2]We also collect an overhead view, as we see in Fig.1, but do not use it for training.

[3]We also collect states, i.e. $\tau = (\mathbf{o}_1, \mathbf{s}_1, r_1, \mathbf{a}_1...)$, although they are not used for policies trained on images.

## 4.2 Visual representation learning

We explore several representation learning methods to pretrain the encoder $\phi$, before policy learning.

**Inverse Dynamics (ID):** An inverse dynamics model estimates the next action from the current observation and a future observation $k$ steps ahead via the objective:

$$\mathcal{L}_{\text{ID}}(\phi, \tau) = \mathbb{E}_{(\mathbf{o}_t, \mathbf{o}_{t+k}, \mathbf{a}_t) \sim \tau} \left\| \mathbf{a}_t - f(\phi(\mathbf{o}_t), \ \phi(\mathbf{o}_{t+k}), \ k) \right\|_2^2. \tag{5}$$

In practice, we replace $\mathbf{o}_t, \mathbf{o}_{t+k}$ with $\tilde{\mathbf{o}}_t, \tilde{\mathbf{o}}_{t+k}$ to combine ID with frame stacking and set $k = 1$.

**Latent Forward Dynamics (LFD):** The latent forward dynamics objective predicts the next latent observation from the current observation and action:

$$\mathcal{L}_{\text{LFD}}(\phi, \tau) = \mathbb{E}_{(\mathbf{o}_t, \mathbf{a}_t, \mathbf{o}_{t+1}) \sim \tau} \left\| \phi^{\text{EMA}}(\mathbf{o}_{t+1}) - g(\phi(\mathbf{o}_t), \ \mathbf{a}_t) \right\|_2^2. \tag{6}$$

To avoid latent collapse [19], we encode $\mathbf{o}_{t+1}$ using a target network $\phi^{\text{EMA}}$, whose weights are an exponential moving average of the weights of $\phi$, with decay rate 0.99.

**AutoEncoder (AE):** An autoencoder [52] jointly trains the encoder $\phi$ with a decoder $\psi$ to minimize the pixel reconstruction loss: $\mathcal{L}_{\text{AE}}(\phi, \tau) = \mathbb{E}_{\mathbf{o}_t \sim \tau} \left\| \psi(\phi(\mathbf{o}_t)) - \mathbf{o}_t \right\|_2^2$.

**Pretrained DINO encoder:** Lastly, we consider a pretrained DINO encoder [10]. Note that we need to pad the $64 \times 64$ observations in the DMC-VB dataset to the $224 \times 224$ size required by DINO.

## 4.3 Baselines using privileged states

The presence of both visual observations and privileged states in DMC-VB enables the evaluation of realistic upper bounds on representation learning methods. We include two state-based baselines.

**BC (state):** This BC variant has access to the states at both training and evaluation time. The actions are predicted as $\hat{\mathbf{a}}_t = \pi(\tilde{\phi}(\mathbf{s}_t))$, where $\tilde{\phi}$ is a MLP.

**State prediction pretraining:** Representations are pretrained to predict states (only accessible during training) via the objective: $\mathcal{L}_{\text{state}}(\phi, \tau) = \mathbb{E}_{(\mathbf{s}_t, \mathbf{o}_t) \sim \tau} \left\| \mathbf{s}_t - h(\phi(\mathbf{o}_t)) \right\|_2^2$, where $h$ is a linear layer.

## 5 Benchmark Experiments

Accompanying DMC-VB, we propose three benchmark evaluations that examine the utility of pretrained visual representations for policy learning in the presence of visual variations. Each benchmark leverages unique properties of DMC-VB—see Table 1—and selects different data subsets to investigate the following questions on visual representation learning for control:

• **(B1)** studies whether visual representation learning makes policies robust to distractors. (Sec. 5.1)

• **(B2)** investigates whether visual representations pretrained on mixed quality data improve policy learning with limited expert data. (Sec. 5.2)

• **(B3)** explores whether visual representations pretrained on tasks with stochastic hidden goals improve policy learning on a new task with fixed hidden goal and limited expert data. (Sec. 5.3)

**Notation:** Following Sec.4, we denote $\mathcal{M}_1 + \mathcal{M}_2$ the agent for which the method $\mathcal{M}_1$ is used to pretrain the encoder $\phi$, and $\mathcal{M}_2$ is used to learn the policy $\pi$. For instance, ID + BC refers to first pretraining the encoder with ID, followed by learning the policy with BC (with a frozen encoder). We denote NULL + BC the agent for which $\phi \circ \pi$ is trained end-to-end with BC. In Secs. 5.2 and 5.3, when we pretrain on a dataset $D_1$, then learn the policy on another dataset $D_2$, we name the agent $\mathcal{M}_1(D_1) + \mathcal{M}_2(D_2)$. In addition, NULL + BC (states) trains end-to-end $\phi \circ \pi$ on states with BC. Lastly, Data refers to the average reward on the training dataset.

**Preprocessing details:** We first normalize actions in $[-1, 1]$ and observations in $[-0.5, 0.5]$. Second, as in [25], we pad each $64 \times 64$ image by 4 pixels on each side, and then apply a random cropping to return a randomly shifted $64 \times 64$ image. Finally, we apply a frame stacking of three consecutive observations. For models which use state, we center and normalize each state dimension.

**Training:** For both representation pretraining and policy learning, we use for 400k Adam iterations on a single NVIDIA A100 GPU with batch size 256 and learning rate 0.001. Appendix D details all the architectures and hyperparameters used.

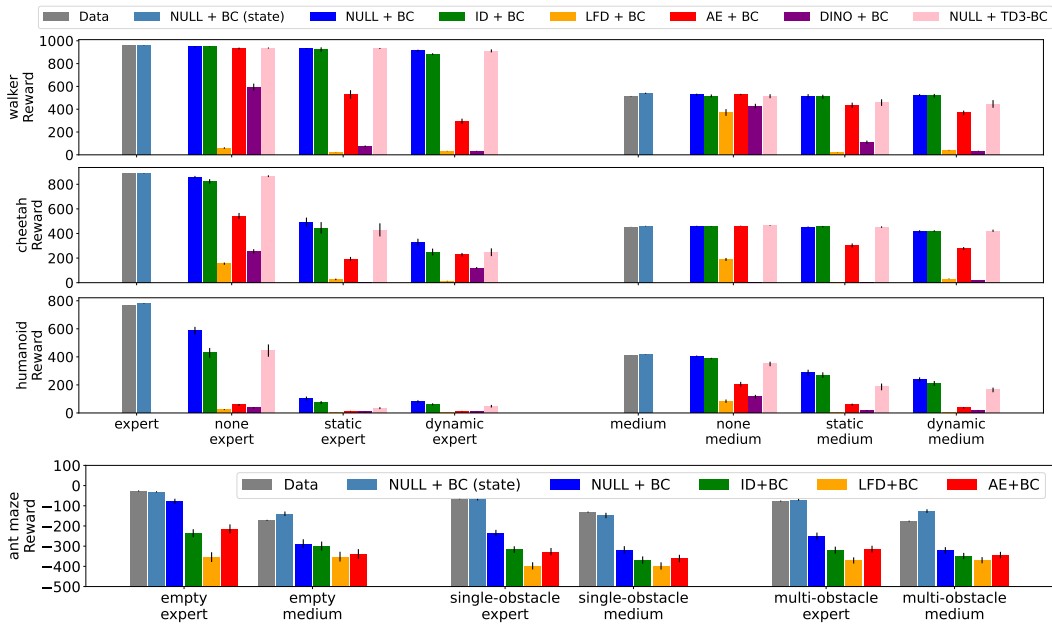

Figure 3: Online evaluation scores on the locomotion tasks [three top rows] and ant maze navigation tasks [bottom row] of DMC-VB, averaged over 30 trajectories, with standard errors. Higher reward is better. For locomotion task rows, results are grouped by distractor type and demonstration data quality. For the ant maze row, results are grouped by maze difficulty and demonstration data quality. NULL + BC is the best overall method. Pretrained representations offer no advantage with or without visual distractors. LFD + BC performs poorly, AE + BC and DINO + BC learn moderate policies, and ID + BC is comparable to NULL + BC. Full scores are in Appendix E.1 and the temporal evolution of rewards through training is plotted in Appendix E.2.

**Online evaluation:** Every 20k training steps, we evaluate the agent online over 30 online rollouts in the evaluation environment. When visual distractors are present, the evaluation environment contains unseen visual distractors from the training distractor distribution. In the figures displayed in this section, we report the best (online) evaluation scores obtained through training.

## 5.1 Policy learning with visual distractors

**Benchmark 1** evaluates the reward collected by different agents on the full DMC-VB expert and medium datasets in Fig.3. For the locomotion tasks, we find that the baseline NULL+BC is the top performing method. ID+BC matches (but does not exceed) this baseline, and the other pretraining methods all perform worse. This indicates that pretrained representations do not seem to help, at least in these benchmark experiments. With distractors, on `walker-expert`, `walker-medium`, and `cheetah-medium`, NULL+BC and ID+BC are the only methods that maintain high rewards suggesting that they learn representations that are robust to visual variations. Lastly, offline RL (NULL+TD3-BC) is outperformed by NULL+BC most of the time, and never exceeds it.

For the DMC-VB navigation tasks, NULL+BC is the best method and pretrained visual representations only harm performance. We observe similar results when plotting the fraction of successes in reaching the goal and the average velocity of the agent toward the goal in Appendix E.6. We observe that for many tasks, performance degrades more strongly with visual distractors for expert than for medium datasets; we expect this is due to the medium dataset having higher coverage (see Fig. 2), reducing the chances of failure by entering a highly out of distribution state.

**Inspecting the visual representations:** To provide insights into the learned visual representations, we freeze the encoder and train two decoders to reconstruct (a) observations, and (b) states (leveraging the states in DMC-VB). Fig. 4 shows that representations that achieve low state reconstruction error capture minimal control-relevant features and lead to better downstream policy performance (ID, BC). Meanwhile, representations that attain low observation reconstruction error are not robust to visual distractors and result in worse downstream policies (LFD, AE, DINO). For locomotion tasks

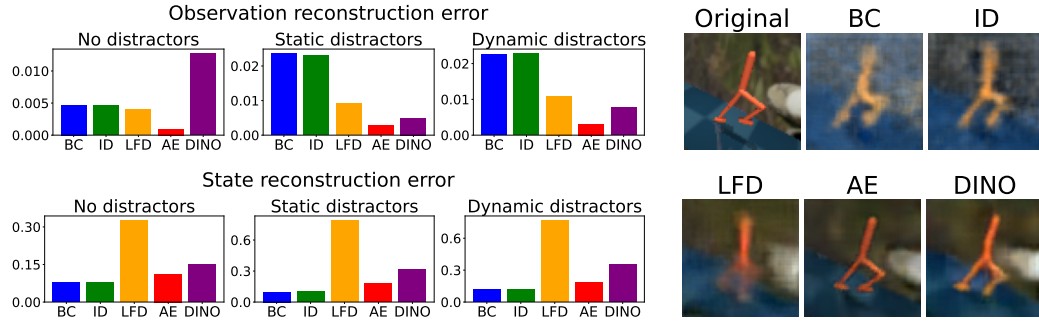

Figure 4: [Left] Least-squared test error for reconstructing the observations [top] and states [bottom], averaged over the different DMC-VB locomotion tasks and policies. Lower is better. As results are averaged over $150k$ samples, the standard errors are too small to be visible. See Appendix E.3, for a detailed breakdown per task and distractor. [Right] Observation reconstruction examples. See Appendix E.4, for additional image reconstructions. BC and ID both (a) discard visual distractors (background object and agent color), and (b) reach the lowest state reconstruction errors.

with visual distractors, both ID and BC achieve the highest observation reconstruction errors and the lowest state reconstruction errors, also discarding control-irrelevant information such as background and agent color. As expected the AE achieves the lowest observation reconstruction errors, and similar to DINO fails to learn good policies due to high state reconstruction errors. Finally, LFD has the highest state reconstruction errors, which explains its bad performance.

**State-based upper bounds:** Policies trained on states are unaffected by the presence of visual distractors, and provide an upper bound for representation learning methods. Fig. 3 shows that the representation gap between policies trained on states versus on observations is minimal in the absence of distractors, but widens significantly in the presence of dynamic distractors. Lastly, we found no advantage in pretraining the encoder to predict the state, which we attribute to the fact that some state components (e.g. velocities) are hard to predict—see Appendix E.5 for details.

## 5.2 Pretraining representations on mixed data helps few-shot policy learning

**Benchmark 2** leverages DMC-VB datasets collected with different behavioral policy levels to investigate whether pretraining a representation on sub-optimal (mixed) data helps policy learning when expert data is scarce. Specifically, for each locomotion task, we first train two vision encoders, using ID and BC on the full *mixed* quality DMC-VB datasets. Second, we freeze the encoders and train the policy networks using only $1\%$ of the *expert* dataset (20 trajectories). Fig. 5 compares these two-stage training agents (in green and red) to a BC agent trained on $1\%$ of the expert dataset (blue), and a BC agent trained on the combined $1\%$ expert dataset and full mixed dataset (orange). The results suggest that (a) limiting expert data impacts performance, (b) merely combining mixed and limited expert data leads to a drop in performance for BC, and (c) pretraining a representation on mixed data provides a performance boost, especially in the presence of visual distractors.

## 5.3 Pretraining on tasks with stochastic hidden goals helps few-shot policy learning

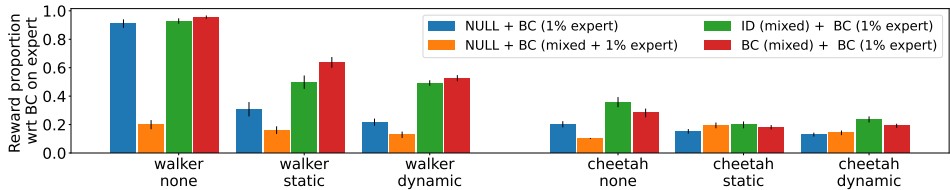

Figure 5: Pretraining encoders on mixed data improves performance when a BC policy is trained on a small expert dataset. BC and ID pretraining perform similarly. For each task, performance is reported as the proportion of reward obtained by BC on full expert data without distractors (higher is better). We include full results including *cheetah*, and LFD/AE pretraining in Appendix F.

**Benchmark 3** considers a variant of the ant maze datasets of DMC-VB, in which image observations are rendered without the goal (green sphere). This setting mimics scenarios in which data is generated by an agent exploring a new environment in an open-ended manner without explicit goals. For each maze, we are given a large *expert* dataset of 1M steps. In each trajectory, the agent moves from a random initial position to a random final position—note the goal cannot be derived from visual observations. We denote this pretraining dataset: *stochastic goals* in Fig. 6. Additionally, for each maze, we have access to five small datasets of ten *expert* trajectories each. All trajectories in each dataset go from a fixed start to a fixed goal location,

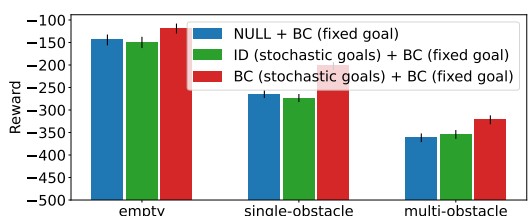

Figure 6: Pretraining representations on tasks with stochastic hidden goals improves performance when subsequently training a policy on tasks with a fixed hidden goal and limited expert data. Higher reward is better. Fraction of successful episodes and average velocity metrics are reported in Appendix G.

with noise added to the agent's body initialization. We denote this dataset: *fixed goal*. As in **(B2)**, Fig. 6 compares BC agents trained directly on the small dataset, to those with two-stage training, for which representations are pre-trained on the large *stochastic goals* dataset. Our findings indicate that pretraining a representation on a collection of tasks with stochastic hidden goals aids policy learning on new tasks with fixed hidden goals. Despite prior findings that ID pretraining on tasks with stochastic hidden goals performs better than BC [7], we find that BC pretraining outperforms ID pretraining on this benchmark.

## 6 Conclusions

We propose DMC-VB, a dataset designed for systematic evaluation of representation learning methods for control tasks in environments with visual distractors. DMC-VB is the first dataset to satisfy six key desiderata, enabling detailed and diverse evaluations. Alongside our dataset, we propose three benchmarks on DMC-VB which compare several established representation methods.

Our results suggest, rather surprisingly, that current pretraining methods (with or without generative losses) do not help BC policy learning on DMC-VB. We leave a detailed investigation into the reason for these results to future work. We also expose a large representation gap between the performance of (a) BC with and without visual distractors, and (b) BC on pixels vs. BC on states, which highlights the need for better representation learning methods. In addition, our findings reveal the potential of leveraging suboptimal datasets, and tasks with stochastic hidden goals, to pretrain representations and learn better policies on DMC-VB with limited expert data.

DMC-VB's unique features present researchers with the tools to investigate fundamental questions in representation learning for control; and to systematically benchmark the performance of future representation learning methods, ultimately contributing to the creation of robust, generalist agents. Our work has two primary limitations. First, DMC-VB could be extended to a broader range of environments, including sparse rewards, multiple agents, complex manipulation tasks, or stochastic dynamics. Second, the synthetic nature of our visual distractors may raise questions about the generalization of our findings to real-world tasks. While our findings may be applicable to robotics tasks, incorporating more diverse and realistic distractors would strengthen our findings.

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

# A  Dataset Documentation

The overall characteristics, properties and intended use of the dataset are laid out in the main paper, predominantly in Section 3. Appendix C also provides further details on the environments and dataset reward distributions.

**License.** The DMC-VB dataset is released under the CC-BY license. The software code to reproduce the benchmark experiments on DMC-VB is released under the Apache 2.0 license.

**Hosting and Maintenance.** The dataset is hosted in a Google Cloud Storage Bucket at: `https://console.cloud.google.com/storage/browser/dmc_vision_benchmark`. The code is hosted on GitHub at: `https://github.com/google-deepmind/dmc_vision_benchmark`. The GitHub page contains instructions to download the dataset using the Google Cloud CLI. Metadata for the dataset is automatically provided via the Google Cloud Storage Bucket. Examples of the dataset and an overview of the dataset structure are provided in the GitHub `README.md`. We, as the dataset owners, will carry out any necessary maintenance to the dataset and software code in order to ensure that it is preserved and accessible in the long term.

**Reading the data.** Detailed instructions for downloading the data are provided in our GitHub page. The GitHub code contains examples of loading the data for our benchmark experiments. The data is loaded using Tensorflow Datasets and the code to load the data can easily be re-used for other experiments.

**Reproducibility.** We have followed the Machine Learning Reproducibility Checklist in order to ensure that all results are easily reproducible. For benchmark experiments, we detail all model architectures and hyperparameters in Appendix D, and the evaluation procedure in the main paper. Scripts to reproduce all experiments in the main paper are provided in our GitHub page. The code release contains all relevant models and evaluation procedures, as well as instructions for accessing the dataset.

**Ethics and Responsible Use.** In terms of the ethical implications of our dataset, AI agents that can learn from offline visual data will eventually be capable of interacting with humans in many domains both physical and virtual, and alignment will be crucial. Our dataset captures simple locomotion and navigation problems with visual variations. Consequently, AI agents trained on this data will have limited capabilities and thus currently minimal ethical considerations.

**Privacy.** Our dataset does not contain any personally identifiable information, and so we do not consider there to be any privacy concerns with our dataset.

**Responsibility Statement.** We, as the dataset owners, confirm that we bear responsibility in case of violation of rights.

# B  Deterministic Exogenous Block Markov Decision Process

A Markov decision process (MDP) [5] is the tuple $\text{MDP} = (\mathcal{S},\ \mathcal{A},\ \mathcal{O},\ T,\ r,\ q(\mathbf{o}|\mathbf{s}))$ where $\mathcal{S}$ is the set of states $\mathbf{s} \in \mathcal{S}$, $\mathcal{A}$ is the set of actions $\mathbf{a} \in \mathcal{S}$, $\mathcal{O}$ is the set of observations $\mathbf{o} \in \mathcal{O}$, $T(\mathbf{s}'|\mathbf{s},\mathbf{a})$ is dynamics function, $r(\mathbf{s},\mathbf{a})$ is the reward function, $q(\mathbf{o}|\mathbf{s})$ is the emission function.

DMC-VB is generated in environments that can represented by Deterministic Exogenous Block MDPs (EX-BMDP) [13]. In this MDP family, the state is factorized into an endogenous latent and an exogenous latent, as shown in Fig.7. The endogenous latent ($\mathbf{s}_1$) captures all factors controllable by the agent while the exogenous latent ($\mathbf{s}_2$) captures all other factors that do not affect the agent or the reward. Additionally, the dynamics are factorized as $T(\mathbf{s}' \mid \mathbf{s}, \mathbf{a}) = T_1(\mathbf{s}_1' \mid \mathbf{s}_1, \mathbf{a})\, T_2(\mathbf{s}_2' \mid \mathbf{s}_2)$, where $T_1$ is deterministic and $T_2$ can be stochastic. The block assumption preserves the full observability of the MDP by ensuring that it is possible to recover $\mathbf{s}_1$ and $\mathbf{s}_2$ from the observation [12, 13]. As the exogenous latent cannot influence the reward, the dashed line in Fig.7 indicates that the optimal action depends only on the endogenous latent.

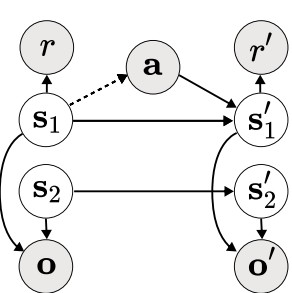

Figure 7: EX-BMDP.

The EX-BMDP framework represents a realistic family of MDPs that appear in many scenarios involving control from visual observations. In the *DM Control Suite* environments considered in our dataset, the endogenous variables are the agent's state (including joint positions, velocities, goal location, etc.), while the exogenous variables are the camera viewpoint, background and agent color. Note that, in our experiments, we apply a frame stacking of three consecutive observations to preserve full observability as the agent's velocity is not computable from a single frame. This guarantees that the block assumption is satisfied.

In robotic tasks, the endogenous variables capture the robot gripper and objects it can interact with while the background and other control-irrelevant objects are exogenous variables. Lastly, in autonomous driving, endogenous variables include all vehicles, agents and obstacles, while background scenery is an exogenous variable.

Despite their simplicity, EX-BMDPs are quite versatile, and can themselves be sub-divided by further factorization of the endogenous state [32].

## C   Dataset Characteristics

### C.1   Reward Distribution

Table 2 presents statistics of the reward distributions for the DMC-VB datasets.

Table 2:   Reward statistics for episodes in the DMC-VB dataset.  Episodes in the locomotion environments are all 500 timesteps while ant maze episodes are variable length as they terminate when the agent reaches the goal. Note rewards are positive in locomotion tasks and negative in ant maze tasks, with higher rewards being always better.

| | Task | Policy level | Steps | Mean | Std. Dev. | Min. | P25 | Median | P75 | Max. |
|---|---|---|---|---|---|---|---|---|---|---|
| Loco-motion Tasks | walker | expert | 1M | 957.8 | 20.4 | 831.9 | 946.8 | 960.1 | 970.9 | 993.0 |
| | | medium | 1M | 513.8 | 108.5 | 9.7 | 503.9 | 534.1 | 563.5 | 694.3 |
| | | mixed | 1M | 177.5 | 179.0 | 10.3 | 48.8 | 86.9 | 255.2 | 622.2 |
| | | random | 1M | 40.1 | 8.2 | 26.6 | 33.7 | 37.9 | 44.9 | 80.9 |
| | cheetah | expert | 1M | 890.5 | 18.5 | 688.8 | 882.3 | 895.1 | 903.3 | 917.8 |
| | | medium | 1M | 454.8 | 26.3 | 86.6 | 446.4 | 457.1 | 468.0 | 495.1 |
| | | mixed | 1M | 284.3 | 132.0 | 33.9 | 151.9 | 336.9 | 391.1 | 465.3 |
| | | random | 1M | 6.8 | 2.6 | 1.0 | 4.9 | 6.5 | 8.4 | 22.5 |
| | humanoid | expert | 1M | 766.0 | 21.3 | 651.4 | 753.7 | 770.7 | 782.5 | 810.8 |
| | | medium | 1M | 410.2 | 15.9 | 10.2 | 402.1 | 410.8 | 420.0 | 449.2 |
| | | mixed | 1M | 139.9 | 137.2 | 0.1 | 5.0 | 60.3 | 272.1 | 372.6 |
| | | random | 1M | 1.1 | 0.8 | 0.0 | 0.5 | 1.0 | 1.6 | 8.2 |
| Ant Maze Tasks | empty | expert | 1M | -26.6 | 20.4 | -160.9 | -39.7 | -22.8 | -10.8 | -0.9 |
| | | medium | 1M | -172.2 | 138.7 | -937.3 | -247.8 | -131.5 | -70.8 | -2.1 |
| | single-obstacle | expert | 1M | -65.5 | 63.6 | -603.4 | -98.7 | -46.1 | -14.3 | -0.7 |
| | | medium | 1M | -131.1 | 185.5 | -1156.9 | -146.3 | -61.0 | -20.2 | -0.7 |
| | multi-obstacle | expert | 1M | -76.6 | 68.2 | -512.1 | -115.6 | -60.7 | -19.1 | -0.9 |
| | | medium | 1M | -176.0 | 181.2 | -1036.7 | -240.6 | -118.7 | -46.8 | -0.8 |

### C.2   State spaces

**Locomotion tasks**. The state space is unmodified from the original DM Control Suite environments and contains only proprioceptive measurements. The state is formed by concatenating the following measurements for each task.

- `walker-walk`: orientations, height, joint velocities (total 24 dimensional).
- `cheetah-run`: joint positions, joint velocities (total 17 dimensional).
- `humanoid-walk`: center of mass velocity, extremity positions, head height, joint angles, torso velocity, joint velocities (total 67 dimensional).

**Ant maze task**. The state space is formed by concatenating the proprioceptive measurements from the DM Control Suite ant maze environment and additional exteroceptive measurements of the ant and goal positions. The original proprioceptive states are: appendage positions, body positions, body quaternions, egocentric target vector, end effector positions, joint positions, joint velocities, sensor accelerometer, sensor gyroscope, and touch sensor. We added the exteroceptive states: ant position, goal position, world-frame vector from the ant to the goal, egocentric vector from the ant to the goal, and shortest path distance from the ant to the goal. The total state space is 167 dimensional.

### C.3   Rewards

**Locomotion tasks**. We use the default reward from the DM Control Suite environment which is 1 when the walker attains a velocity greater than the target velocity and is zero otherwise.

**Ant maze**. We define our own dense reward as:

$$r = -d/d_{max} \tag{7}$$

where $d$ is the current shortest path distance from the ant to the target, and $d_{max}$ is the maximum shortest path distance to the target from any point in the maze. We use the Fast Marching Method

algorithm [43] to compute the shortest path lengths in the maze. Episodes terminate if and when the ant makes contact with the target.

## C.4   Further details for ant maze dataset

**Behavioral policy training:**   We train one behavioral policy per maze layout (per task) which learns to navigate from arbitrary start to goal positions. The behavioral policy takes as input the proprioceptive states, the ant position, the target position, the world-frame vector from ant to target, and the egocentric vector from ant to target. The maze layout is not part of the state space and the agent must learn the maze layout from the reward. Ant and target positions are initialized randomly at the start of each episode.

**Observations:**   During generation, we collect four observations for each timestep: (a) a top-down view of the maze, (b) an egocentric view, (c) a follow-up view–which is useful to infer the proprioceptive states—(d) an overhead view. See Fig.1. We additionally collect these same observations by hiding the goal. Note that we do not use the overhead view for training.

# D Architecture and hyperparameters

Table 3 summarizes all the different architectures and hyperparameters used in our experiments. We detail some choices below.

## D.1 Encoders

We use a simple observation encoder architecture, similar to Brandfonbrener et al. [7]. The only difference we make is that similar to [33], we use a frame stacking of 3. The stacked RGBs input, of size $64 \times 64 \times 9$ (for locomotion tasks) or $64 \times 64 \times 27$ (for ant-maze tasks, since we have three images at each timestep) go through a convolutional neural network (CNN) with four layers. The CNN layers use filters of size $3 \times 3$, strides of $(2, 2, 1, 1)$, gaussian error linear units (gelu) [24] activations, and an increasing number of $(32, 64, 128, 256)$ channels. The CNN output, of size $16 \times 16 \times 256$ is flattened into a $65,536$-dimensional vector before being mapped to a low dimensional vector of size $64$ via a linear *trunk* layer (which is the layer with the largest number of parameters of the model). Finally, the output goes through a normalization layer [3] and a hyperbolic tangent activation function.

For TD3-BC, we use separate actor trunk layer and critic trunk layers. For good performance, we found it important to use the actor loss to update (a) the CNN shared by the actor and critic encoder, (b) the actor trunk layer, and (c) the critic trunk layer.

For the BC agent trained on states, we do not use frame stacking or the CNN. The state encoder consists of the linear trunk layer, followed by the normalization layer and a `tanh` activation.

## D.2 Default MLP module

The default MLP modules in our experiments consist of a two hidden layers MLP, each of hidden size 256, with rectified linear unit (relu) activation functions. Since all the actions in DMC-VB are normalized to be between $-1$ and $1$, we apply a hyperbolic tangent activation to the output.

## D.3 Decoders

Our decoder network draws inspiration from [52] and reverts the visual encoder. First, we map the 64-dimensional observation encoding back to a $65,536$-dimensional vector, which is then unflattened into a $16 \times 16 \times 256$ tensor. This tensor goes through a deconvolutional neural network (DeCNN), with filters of size $3 \times 3$ filters, strides of $(1, 1, 2, 2)$, gaussian error linear units (gelu) [24], and a decreasing number of channels of $(128, 64, 32, 9)$. We do not use activation for the last layer. This last output, of dimension $64 \times 64 \times 9$, is then unflattened into three images (for locomotion tasks) or nine images (for ant-maze tasks), each image being of size $64 \times 64 \times 3$.

The state reconstruction module used in **(B1)** is a default MLP. However, we do not use activation for the last layer, which predicts the state.

## D.4 Agent details

This section details the different agents used.

**BC** For BC, the action prediction network is the default MLP above.

**ID** With the notations of Equation 5, $f$ concatenates the embeddings $\phi(\mathbf{o}_t)$, $\phi(\mathbf{o}_{t+k})$, and (if present) the embedding of $k$ and passes them the inverse dynamics action prediction network, which is the default MLP above. Our code supports setting the number $k$ of ID steps to a fixed value, or sampling uniformly in the range $\{1, \ldots, k_{\max}\}$ at each iteration. For the latter, we use a linear layer to map $k$ to a 64-dimensional embedding.

**LFD** With the notations of Equation 6, $g$ concatenates $\phi(\mathbf{o}_t)$ and an embedding of $\mathbf{a}_t$, before passing them through a latent forward predictor network, The inverse dynamics action prediction network, which is the default MLP above. As detailed in Equation 6, we encode the future observation using a target network $\phi^{\text{EMA}}$, which has the same architecture as the encoder network $\phi$ and whose

Table 3: Hyperparameters and architecture used for the different agents evaluated on DMC-VB

| Module | Hyperparameter | Value |
|---|---|---|
| Observation encoder | Frame stacking | 3 |
| | CNN number of channels | $(32, 64, 128, 256)$ |
| | CNN kernel sizes | $(3, 3, 3, 3)$ |
| | CNN activation | gelu |
| | CNN kernel strides | $(2, 2, 1, 1)$ |
| | Output activation | tanh |
| | Output dimension | 64 |
| State encoder | Frame stacking | 1 |
| | Output activation | tanh |
| | Output dimension | 64 |
| State predictor layer | Output dimension | Equal to state dimension |
| Action predictor | Hidden layer sizes | $(256, 256)$ |
| ID action predictor | Activation function | relu |
| | Output activation | tanh |
| | Output dimension | Equal to action dimension |
| LFD predictor | Hidden layer sizes | $(256, 256)$ |
| Action encoder | Activation function | relu |
| | Output activation | tanh |
| | Output dimension | 64 |
| | LFD target network decay rate | 0.99 |
| ID step embedding | Output dimension | 64 |
| | Number of ID steps | 1    (default) |
| | Sample ID steps | False    (default) |
| Actor, Critic 1 and Critic 2 | Hidden layer sizes | $(256, 256)$ |
| | Activation function | relu |
| | Last layer activation | tanh |
| | Output dimension | 64 |
| TD3-BC | Trade-off $\alpha$ | 2.5 |
| | Discount factor | 0.99 |
| | Policy noise | 0.2 |
| | Policy noise clipping | 0.5 |
| | Policy update frequency | 2 |
| | Target network decay rate | 0.99 |
| | Loss for shared CNN | actor loss |
| | Loss for actor/critic trunks | actor loss |
| Observation decoder | Linear layer dimension | $16 \times 16 \times 256$ |
| | DeCNN number of channels | $(128, 64, 32, 9)$ |
| | DeCNN kernel sizes | $(3, 3, 3, 3)$ |
| | DeCNN activation | gelu |
| | DeCNN kernel strides | $(1, 1, 2, 2)$ |
| | Output activation | Not used |
| | Output shape | Equal to observation shape |
| State decoder | Hidden layer sizes | $(256, 256)$ |
| | Activation function | relu |
| | Output activation | Not used |
| | Output dimension | Equal to state dimension |
| Learning | Number of iterations | $400,000$ |
| | Batch size | 256 |
| | Optimizer | Adam |
| | Learning rate | 0.001 |
| | Online evaluation frequency | $20,000$ |
| Online evaluation | Number of runs | 30 |
| | Action repeat | 2 |

weights $\mu^{\text{EMA}}$ are an exponential moving average of the weights of the teacher network $\mu$. At each gradient update, we set $\mu^{\text{EMA}} \leftarrow \beta\mu^{\text{EMA}} + (1 - \beta)\mu$, where $\beta$ is the decay rate, which we fix to $0.99$.

**AE**   We use the decoder described above.

**TD3-BC**   With the notations of Equation 4, on each mini-batch $\mathcal{B}$, in order to be agnostic to the scalar of the reward, we set $\lambda = \frac{\alpha}{\frac{1}{|\mathcal{B}|}\sum_{\mathbf{o}\in\mathcal{B}}|Q_1(\phi(\mathbf{o}),\pi(\phi(\mathbf{o}))|}$ with $\beta = 2.5$. Here, TD3-BC takes as input (a stack) of images. The actor and the two critic networks are the default MLP modules above. We use a decay rate of $0.99$ for the target actor and critic networks—which are used to compute the target for both critics. Note that, as in [16], the actor network and the target networks are updated twice less frequently than the critics.

For good performance, we found it important to use (a) a separate linear trunk layer for the actor and the critic, and (b) the actor loss to update the shared CNN and the actor and critic trunk layers. This guarantees that for $\alpha = 0$, we recover BC.

# E  Additional results for Benchmark 1

## E.1  Table of results for Benchmark 1

We first present the table of results accompanying our summary Fig.3.

Table 4: Reward mean and standard error for all methods on each of the locomotion datasets. We underline all rewards within 5% of BC (state), which represents an upper bound on the learned representation. Best scores for each dataset are bolded. For random data we do not bold or highlight any scores as all rewards are comparably low.

| | Policy level | Distractor | No pretraining | | Pretraining + BC | | | | State privileged info | | |
| --- | --- | --- | --- | --- | --- | --- | --- | --- | --- | --- | --- |
| | | | NULL + BC | NULL + TD3-BC | ID + BC | LFD + BC | DINO + BC | AE + BC | State + BC | BC (state) | Data |
| walker | expert | none | 948 ± 4 | **950 ± 68** | 947 ± 3 | 59 ± 7 | 594 ± 31 | 932 ± 8 | 954 ± 3 | 960 ± 2 | 958 ± 0 |
| | | dynamic | **913 ± 10** | 881 ± 55 | 883 ± 8 | 33 ± 2 | 33 ± 2 | 294 ± 23 | 886 ± 8 | | |
| | | static | **932 ± 4** | 891 ± 73 | 924 ± 19 | 23 ± 1 | 75 ± 7 | 529 ± 38 | 927 ± 6 | | |
| | expert + medium | none | 741 ± 24 | **797 ± 56** | 697 ± 25 | 71 ± 6 | 445 ± 21 | 684 ± 24 | 727 ± 23 | 784 ± 19 | 736 ± 4 |
| | | dynamic | 621 ± 24 | 558 ± 48 | **638 ± 30** | 33 ± 2 | 35 ± 2 | 410 ± 20 | 608 ± 27 | | |
| | | static | 725 ± 29 | 671 ± 56 | 634 ± 25 | 25 ± 1 | 28 ± 1 | 422 ± 25 | 666 ± 23 | | |
| | medium | none | **529 ± 8** | 502 ± 41 | 517 ± 13 | 371 ± 29 | 428 ± 20 | 529 ± 6 | 540 ± 7 | 539 ± 6 | 514 ± 2 |
| | | dynamic | **524 ± 9** | 461 ± 40 | 518 ± 16 | 41 ± 2 | 33 ± 3 | 369 ± 19 | 492 ± 19 | | |
| | | static | **513 ± 19** | 455 ± 41 | 509 ± 18 | 24 ± 1 | 111 ± 14 | 434 ± 24 | 512 ± 13 | | |
| | mixed | none | 202 ± 32 | 233 ± 38 | **239 ± 31** | 89 ± 7 | 30 ± 2 | 230 ± 26 | 244 ± 31 | 298 ± 27 | 178 ± 4 |
| | | dynamic | 205 ± 29 | **216 ± 32** | 190 ± 27 | 46 ± 2 | 37 ± 2 | 134 ± 15 | 218 ± 26 | | |
| | | static | 178 ± 28 | 182 ± 31 | **212 ± 29** | 37 ± 2 | 69 ± 8 | 134 ± 15 | 214 ± 28 | | |
| | random | none | 24 ± 2 | 23 ± 1 | 22 ± 2 | 21 ± 1 | 30 ± 2 | 21 ± 1 | 29 ± 2 | 38 ± 3 | 40 ± 0 |
| | | dynamic | 28 ± 2 | 24 ± 2 | 22 ± 2 | 21 ± 1 | 31 ± 2 | 22 ± 1 | 31 ± 2 | | |
| | | static | 22 ± 2 | 24 ± 2 | 27 ± 2 | 21 ± 1 | 29 ± 3 | 21 ± 1 | 31 ± 2 | | |
| cheetah | expert | none | 861 ± 7 | **866 ± 54** | 824 ± 18 | 155 ± 9 | 254 ± 18 | 543 ± 24 | 888 ± 3 | 890 ± 2 | 890 ± 0 |
| | | dynamic | **329 ± 28** | 288 ± 32 | 250 ± 27 | 13 ± 2 | 120 ± 7 | 228 ± 14 | 326 ± 29 | | |
| | | static | **493 ± 36** | 445 ± 51 | 447 ± 46 | 27 ± 6 | 3 ± 0 | 194 ± 15 | 492 ± 35 | | |
| | expert + medium | none | **692 ± 25** | 630 ± 37 | 608 ± 25 | 130 ± 6 | 4 ± 0 | 461 ± 2 | 697 ± 25 | 837 ± 17 | 697 ± 4 |
| | | dynamic | **431 ± 7** | 401 ± 26 | 431 ± 4 | 15 ± 1 | 76 ± 5 | 265 ± 13 | 421 ± 8 | | |
| | | static | **471 ± 18** | 441 ± 20 | 454 ± 3 | 1 ± 0 | 3 ± 0 | 321 ± 13 | 459 ± 3 | | |
| | medium | none | 460 ± 2 | **462 ± 3** | 458 ± 2 | 189 ± 12 | 4 ± 0 | 462 ± 2 | 462 ± 2 | 460 ± 2 | 455 ± 1 |
| | | dynamic | 417 ± 10 | **421 ± 22** | 420 ± 6 | 32 ± 2 | 17 ± 1 | 278 ± 12 | 431 ± 4 | | |
| | | static | 454 ± 3 | 368 ± 16 | **455 ± 6** | 1 ± 0 | 3 ± 0 | 303 ± 16 | 456 ± 3 | | |
| | mixed | none | **126 ± 6** | 1 ± 0 | 107 ± 5 | 9 ± 0 | 4 ± 0 | 104 ± 6 | 119 ± 7 | 409 ± 3 | 303 ± 3 |
| | | dynamic | **236 ± 16** | 1 ± 0 | 143 ± 12 | 2 ± 0 | 17 ± 1 | 188 ± 8 | 156 ± 13 | | |
| | | static | 217 ± 18 | 1 ± 0 | **235 ± 18** | 1 ± 0 | 2 ± 0 | 222 ± 13 | 174 ± 13 | | |
| | random | none | 2 ± 0 | 1 ± 0 | 1 ± 0 | 0 ± 0 | 4 ± 0 | 0 ± 0 | 1 ± 0 | 1 ± 0 | 7 ± 0 |
| | | dynamic | 5 ± 0 | 2 ± 0 | 2 ± 0 | 0 ± 0 | 18 ± 1 | 0 ± 0 | 3 ± 0 | | |
| | | static | 1 ± 0 | 0 ± 0 | 2 ± 0 | 0 ± 0 | 2 ± 0 | 0 ± 0 | 1 ± 0 | | |
| humanoid | expert | none | **587 ± 27** | 364 ± 50 | 428 ± 35 | 24 ± 3 | 36 ± 5 | 57 ± 6 | 116 ± 14 | 780 ± 2 | 766 ± 0 |
| | | dynamic | **82 ± 9** | 29 ± 5 | 61 ± 8 | 2 ± 0 | 9 ± 1 | 13 ± 2 | 45 ± 5 | | |
| | | static | **103 ± 14** | 37 ± 7 | 74 ± 10 | 1 ± 0 | 8 ± 1 | 14 ± 2 | 47 ± 6 | | |
| | expert + medium | none | **417 ± 9** | 274 ± 27 | 372 ± 12 | 30 ± 4 | 83 ± 9 | 72 ± 9 | 308 ± 16 | 585 ± 21 | 588 ± 3 |
| | | dynamic | **190 ± 17** | 99 ± 16 | 184 ± 16 | 2 ± 0 | 10 ± 2 | 35 ± 4 | 94 ± 10 | | |
| | | static | **257 ± 19** | 177 ± 24 | 245 ± 16 | 2 ± 0 | 9 ± 2 | 39 ± 6 | 151 ± 17 | | |
| | medium | none | **406 ± 3** | 339 ± 27 | 388 ± 5 | 84 ± 12 | 117 ± 13 | 205 ± 17 | 343 ± 10 | 416 ± 1 | 410 ± 0 |
| | | dynamic | **238 ± 16** | 147 ± 19 | 211 ± 17 | 2 ± 0 | 16 ± 2 | 37 ± 5 | 132 ± 15 | | |
| | | static | **288 ± 21** | 185 ± 27 | 270 ± 19 | 2 ± 1 | 17 ± 3 | 59 ± 8 | 172 ± 19 | | |
| | mixed | none | **152 ± 16** | 20 ± 4 | 145 ± 15 | 2 ± 0 | 19 ± 4 | 58 ± 8 | 158 ± 14 | 227 ± 10 | 140 ± 3 |
| | | dynamic | **46 ± 8** | 10 ± 2 | 21 ± 4 | 2 ± 0 | 3 ± 1 | 6 ± 1 | 20 ± 4 | | |
| | | static | **97 ± 15** | 2 ± 1 | 58 ± 10 | 2 ± 0 | 2 ± 0 | 8 ± 1 | 55 ± 10 | | |
| | random | none | 1 ± 0 | 1 ± 0 | 1 ± 0 | 1 ± 0 | 1 ± 0 | 1 ± 0 | 1 ± 0 | 1 ± 0 | 1 ± 0 |
| | | dynamic | 1 ± 0 | 1 ± 0 | 1 ± 0 | 1 ± 0 | 2 ± 0 | 1 ± 0 | 1 ± 0 | | |
| | | static | 1 ± 0 | 1 ± 0 | 1 ± 0 | 1 ± 0 | 1 ± 0 | 1 ± 0 | 1 ± 0 | | |

## E.2    Time series of rewards

As indicated in the main text, we evaluate each method online every 20k training steps. Fig.8 displays the temporal evolution of the average rewards, when trained on the full expert datasets for locomotion tasks. As expected, we find that training on state information converges very fast and that humanoid training can take a large number of steps due to the difficulty of the task.

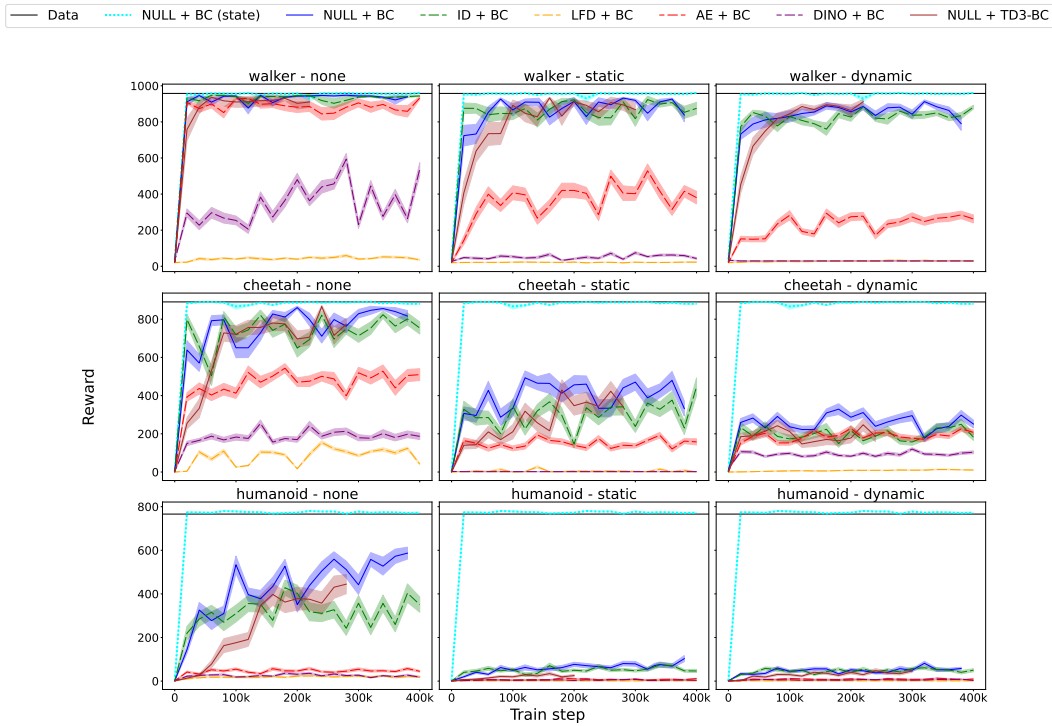

Figure 8:   Time-series of rewards scores on the locomotion tasks of DMC-VB, averaged over 30 trajectories, with standard errors represented as the shaded area. These are all trained on expert trajectories. Higher reward is better.

### E.3 Observations and states reconstruction errors on locomotion datasets

Figures 9 and 10 display the least-squared test errors for respectively reconstructing the observations and the states learned by a decoder with frozen encoder. Each figure includes plots for both expert and medium datasets for all locomotion task. These figures breakdown the summarized plots in the main paper which are averaged over all datasets.

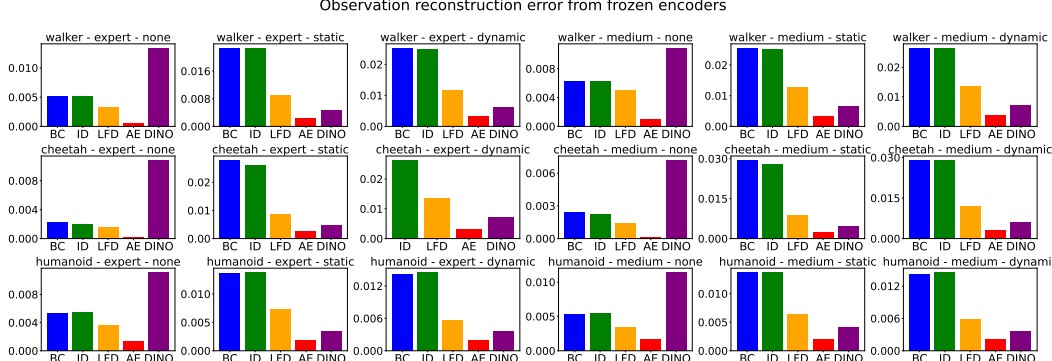

Figure 9: Observations reconstruction errors for the different representation learning methods. The observation decoder is trained by freezing the encoder.

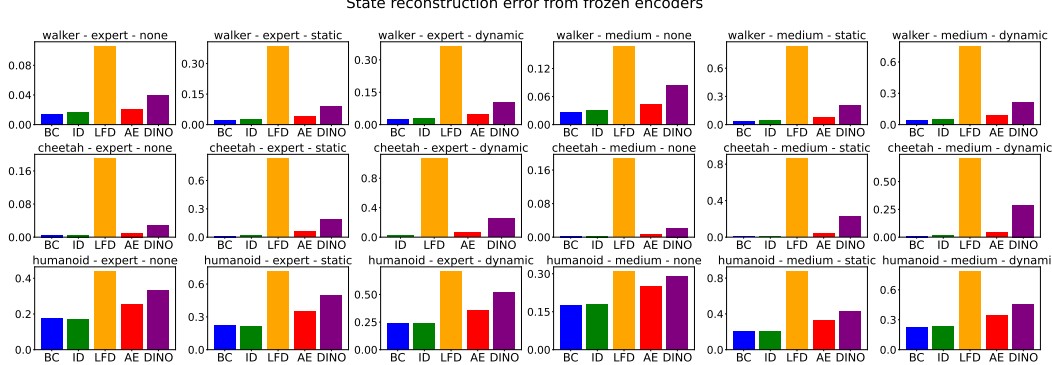

Figure 10: State reconstruction errors for the different representation learning methods. The state decoder is trained by freezing the encoder.

## E.4 Observations reconstructions examples

Fig. 11 presents some observations reconstructed by the different representation methods, on the `walker-walk` tasks, with dynamic distractors.

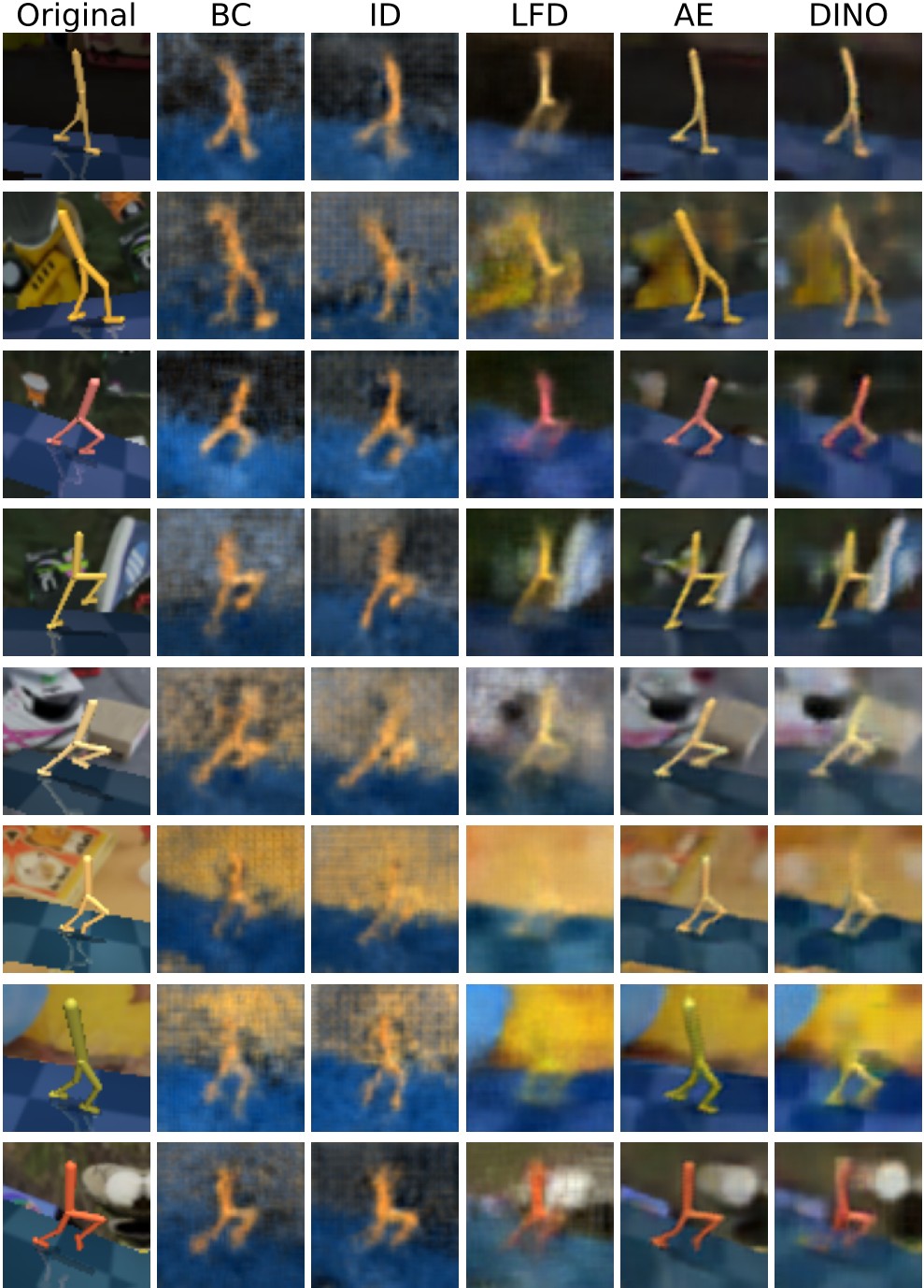

Figure 11: Observation reconstruction from a decoder with frozen encoder, for the different representation learning methods.

## E.5 State prediction on locomotion datasets

For the method State + BC, during pretraining, a visual encoder is trained to directly regress the state from the corresponding observation. In Fig.12, we show the $\ell_1$ state prediction error on held out evaluation data for the locomotion tasks. The error is broken down into errors for each part of the state space. Note that, as we have centered and normalized each state dimension during preprocessing, the errors for the different parts are comparable.

Despite frame stacking, we find that velocity is the most challenging to predict. Additionally, we see that the state for `humanoid` is much harder to predict than for `walker` and `cheetah` due to the high degree of freedom body.

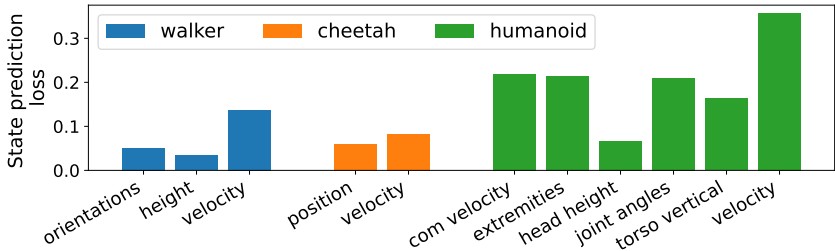

Figure 12: Absolute error evaluation loss for state prediction on the locomotion datasets, when regressed directly from the corresponding image observation.

## E.6 Additional metrics on ant maze environments

For ant-maze environments in **(B1)**, we use the total reward per episode to compare the different agents in Sec.5.1. Here we report two additional metrics: (a) the fraction of success in reaching the goal over the 30 online evaluation rollouts (Fig. 13), and (b) for trajectories reaching the goal, the average velocity towards the goal (Fig. 14). These are useful measures of an agent's behavior in the environment that are independent of the initial distance between the agent and the goal, unlike the reward.

These metrics highlight the gap between the LFD + BC agent and other agents. The LFD + BC agent never manages to reach the goal, despite often attaining reasonable scores in the reward.

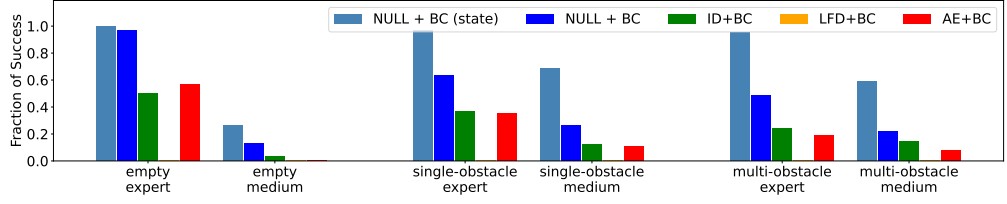

Figure 13: Fraction of success in reaching the goal during the online evaluation of different benchmark models in Ant maze. As in the main paper, NULL+BC achieves the highest fraction of success and pretrained visual representations only harm performance.

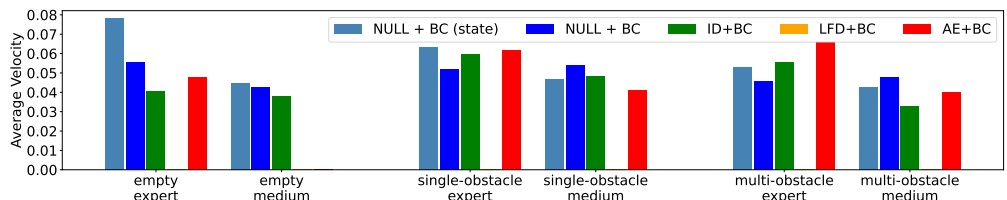

Figure 14: Average velocity of the agent towards the goal during online evaluation of different benchmark models in Ant maze. Velocity is zero if the agent does not succeed in reaching the goal.

# F    Additional results for Benchmark 2

In addition to the inverse dynamics (ID) and the behavior cloning (BC) methods presented in Fig. 5, we evaluate latent forward dynamics (LFD) and autoencoder (AE) for representation learning on the mixed dataset. We present extended results including those additional methods in Fig. 15.

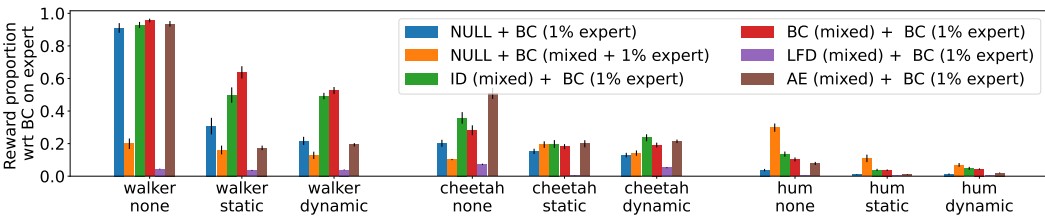

Figure 15: Pretraining encoders on mixed data using different representation learning objectives improve performance when BC policy is trained on small expert data. Performance using different pretraining objectives is shown as the proportion of reward obtained by BC on full expert data without distractors for each task.

# G    Additional metrics for Benchmark 3

Similar to Benchmark 1 above, we present the fraction of success and the average velocity metrics for ant maze for Benchmark 3. Fig. 16 shows the fraction of success and average agent velocity metrics respectively for Benchmark 3. These metrics are consistent with the reward reported in Fig. 6, in showing that BC pretraining on stochastic hidden goal tasks helps few-shot policy learning.

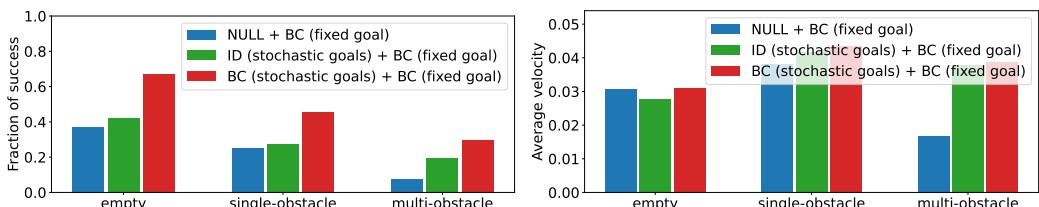

Figure 16: Fraction of success and average velocity of the agent towards the goal during online evaluation of Benchmark 3. Average velocity is only computed for trajectories that reach the goal and is zero otherwise.

## H Frame stacking ablation on locomotion datasets

Fig. 17 compares BC with frame stacking of three consecutive frames versus without frame stacking on DMC-VB expert datasets for locomotion tasks. We see that frame stacking is crucial for learning, likely due to the fact that the agent's velocity is not inferable from a single observation. As a consequence of this result, we use frame stacking in all experiments.

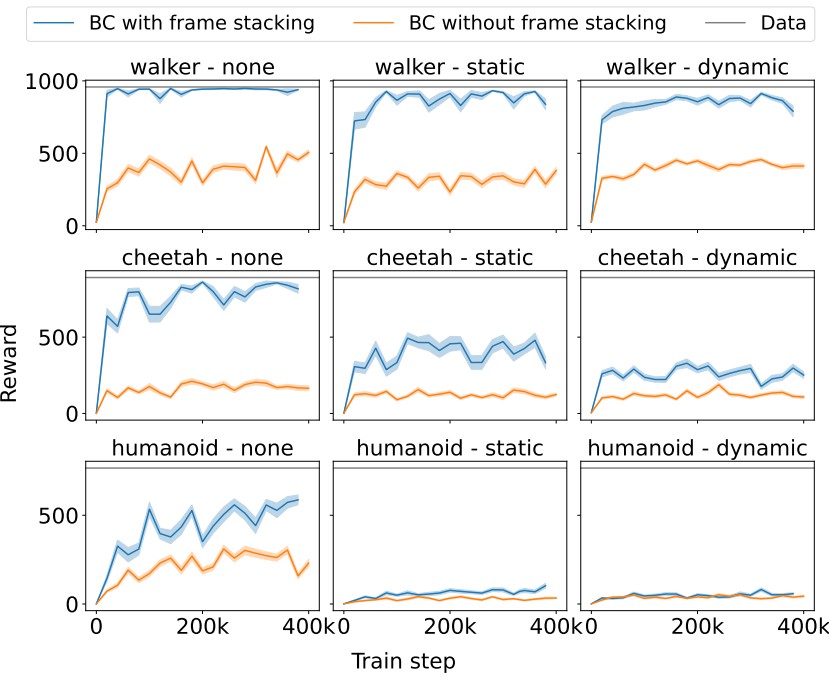

Figure 17: Frame stacking significantly boosts BC performance on DMC-VB.

## I Dataset size study

To validate the choice of making DMC-VB a large dataset with 1M steps per data subset, we evaluate the performance of behavioral cloning on smaller percentages of the dataset. We see in Fig.18 that for harder tasks, and in the presence of visual distractors, BC agents benefit from large training datasets. This suggests that *VD4RL* [33], which only contains 100k steps per dataset, is not sufficiently large to evaluate visual representation learning for control, and this motivates our decision to make DMC-VB 10× larger.

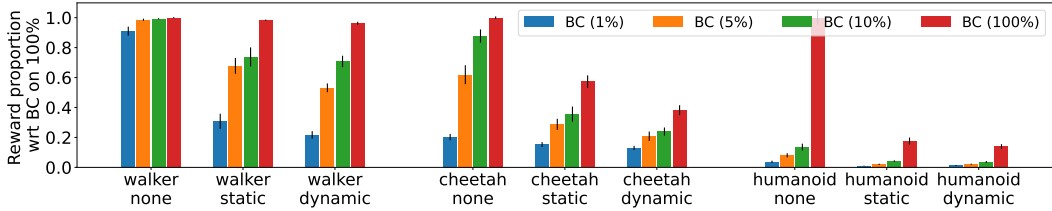

Figure 18: Performance of behavioral cloning without pretraining (NULL+BC) on different percentages of the expert dataset for the DMC-VB locomotion tasks. Rewards are plotted as a proportion of the reward when using 100% of the distractor-free (*none*) dataset. Scores significantly drop when using smaller proportions of the dataset.

## J Multistep Inverse Dynamics Pretraining

In experiments in the main paper, we found that single-step inverse dynamics pretraining is consistently on par or worse than the simple behavioral cloning baseline. Here we evaluate multistep inverse dynamics pretraining corresponding to different values of $k$ in Equation 5. Figure 19 shows that as the value of $k$ increases, the performance of ID pretraining converges towards the performance of behavioural cloning. For larger $k$ the future frame becomes less useful for predicting the current action and the ID objective becomes equivalent to behavioral cloning. The figure also includes the BC + BC baseline, in which pretraining and policy learning share the same objective and data. This is equivalent to NULL + BC and ID + BC as $k$ becomes large.

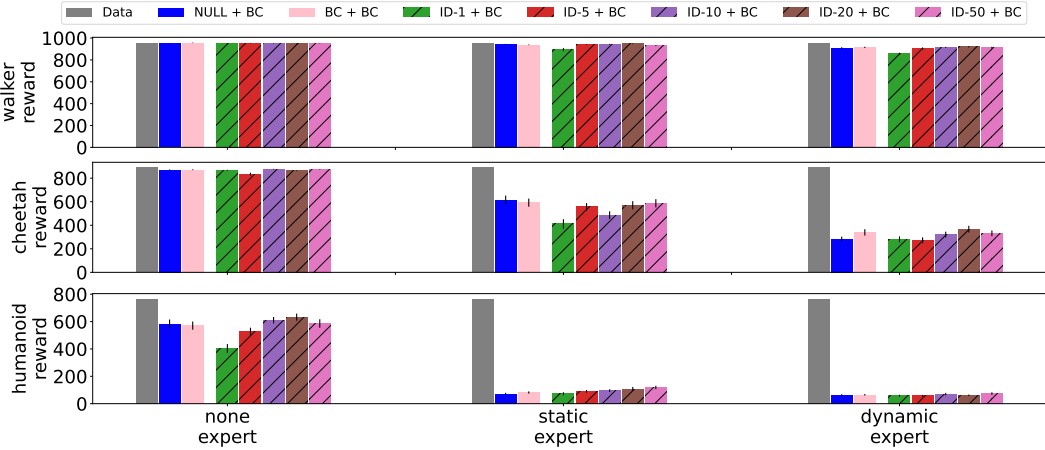

Figure 19: k-step inverse dynamics pretraining is denoted ID-k. Performance improves for larger values of k and tends towards the performance of BC pretraining.

