# OpenReview forum: "DMC-VB: A Benchmark for Representation Learning for Control with Visual Distractors"
_NeurIPS.cc/2024/Datasets_and_Benchmarks_Track — NeurIPS 2024 Track Datasets and Benchmarks Poster_

### Official Review · Reviewer_RVem · 2024-07-24
**Useful contribution to the decision-making community**

**Rating:** 6
**Confidence:** 3
**Correctness:** Yes.
**Clarity:** Yes.

**Review:**

- Overall, this paper provides a commendable benchmark for representation RL with visual distractors.
- I have a question about Sec5.1 that “the other pretraining methods all perform worse”. This is very counterintuitive. Could you provide more about the details of pretraining and have you tried other more powerful pretrained models?

**Strengths:**

- It covered a broad range of the different environments: both vision and state observations, locomotion and navigation tasks of varying difﬁculties.
- The comparisons done across different settings are quite interesting. Some findings are non-intuitive and insightful。
- The authors provide datasets, and code for training and evaluating their proposed method. This significantly facilitates reproducing the experimental results and extending the introduced method.

**Additional Feedback:**

N/A.

**Documentation:**

Github repository has pretty good documentation and it was easy for me to run some of their experiments.

**Ethics:**

I don't see any ethical concerns.

**Limitations:**

I don't see any negative societal impact.

**Opportunities For Improvement:**

- In the benchmarks, RL algorithms are mainly based on BC. I encourage the authors to include comparisons with some offline RL algorithms.
- It would be beneficial to include comparisons with more complex environments like DMC-VB in the dataset and experimental comparisons.

**Relation To Prior Work:**

Yes.

**Summary And Contributions:**

The paper proposes DMC-VB, a dataset designed for systematic evaluation of representation learning methods for control tasks in environments with visual distractors. And the authors propose three benchmark evaluations that examine the utility of pretrained visual representations for policy learning in the presence of visual variations.

---

> ### Author Rebuttal · Authors · 2024-08-15
>
> Thank you for your positive comments and helpful feedback.
>
> > I have a question about Sec5.1 that “the other pretraining methods all perform worse”. This is very counterintuitive.
>
> We also found this result surprising and counterintuitive. To provide intuition towards this result, Figure 4 inspects each learned representation by evaluating its ability to decode the state or pixel observations from the encoding.
> We see that, although the autoencoder and DINO learn good features for pixel reconstruction, their features are not as predictive of the agent state (which is crucial for locomotion) as the BC or ID features. The autoencoder and DINO encodings also capture visual distractors such as the agent’s body color and the background, while the BC and ID features are robust to these visual distractors.
> These results shed light on why pretraining representations can perform worse than the simple BC baseline.
>
> > Could you provide more about the details of pretraining and have you tried other more powerful pretrained models?
>
> As presented in Section 4, in the pretraining phase the visual encoder is optimized to minimize a representation learning loss (Equation 1). The various representation losses that we consider are described in Section 4.2. The dataset used for pretraining the encoder depends on the particular benchmark. For example, in Benchmark 1, we use the same dataset for policy learning and for pretraining the visual encoder.
> DINO is the only large/powerful pretrained model that we tried; it represents a different architecture family to the other CNNs, being a vision transformer.
>
> > In the benchmarks, RL algorithms are mainly based on BC. I encourage the authors to include comparisons with some offline RL algorithms.
>
> We consider the offline RL method TD3-BC ([https://arxiv.org/pdf/2106.06860](https://arxiv.org/pdf/2106.06860)) in our benchmark evaluations. Results for TD3-BC are presented in Figures 3 and 8, as well as in Table 4, Appendix E.1. As TD3-BC was outperformed by BC, we did not include TD3-BC in all experiments, only using BC for clarity of presentation. TD3-BC is implemented in our open-source code [here](https://github.com/google-deepmind/dmc_vision_benchmark/blob/main/dmc_vision_benchmark/rep_learn/td3_bc.py).
>
> > It would be beneficial to include comparisons with more complex environments like DMC-VB in the dataset and experimental comparisons.
>
> In the related work (Section 2), we discuss environments (lines 71-78) and offline datasets (lines 79-93) that relate to DMC-VB. Further, in Table 1 we compare the 5 most closely related prior datasets to DMC-VB across six key properties that we have identified.
>
> We emphasize that we are not presenting a novel method but rather a novel dataset and benchmark which enables the evaluation of recent representation learning methods. Evaluating representation learning methods on prior datasets is out of the scope of our paper, and indeed prior investigations have been unable to examine the same questions due to the limitations of existing datasets (as discussed in Section 2).

---

### Official Review · Reviewer_n4Ln · 2024-07-25
**Review of the paper**

**Rating:** 8
**Confidence:** 3
**Correctness:** The claims made in the submission is …
**Clarity:** Yes, the paper is well written.

**Review:**

The DeepMind Control Visual Benchmark (DMC-VB) paper significantly advances the field of offline RL by introducing a comprehensive dataset and robust benchmarks for evaluating representation learning methods. While it addresses critical gaps in existing research, particularly in visual domain generalization, it also highlights ongoing challenges that require continued exploration and innovation. The paper's contribution to promoting transparency, reproducibility, and methodological rigor in RL research makes it a valuable addition to the literature.

**Strengths:**

1, DMC-VB covers a wide range of tasks, including locomotion and navigation, with varying levels of difficulty. It introduces static and dynamic visual variations, ensuring comprehensive evaluation of agent robustness.

2, The paper innovates by systematically pairing state and pixel observations, enabling a thorough investigation into the "representation gap" between these approaches.

3, The proposed benchmarks provide clear metrics for evaluating representation learning methods, contributing to standardization and advancement in the field.

**Additional Feedback:**

None

**Documentation:**

Yes, it is sufficient detail on Documentation.

**Limitations:**

The same as the Opportunities For Improvement.

**Opportunities For Improvement:**

1, Lack of the complex control tasks limit the quality of this work. Maybe you can try some robotics task?
2, Fewer Visual RL baselines.

**Relation To Prior Work:**

The paper clearly discussed the related work.

**Summary And Contributions:**

The paper introduces the DeepMind Control Visual Benchmark (DMC-VB), a dataset designed to evaluate the robustness of offline reinforcement learning (RL) agents in solving continuous control tasks from visual inputs amidst various visual distractions. This dataset addresses shortcomings observed in previous works by incorporating diverse tasks, static and dynamic visual variations, and policies with varying skill levels. It provides both pixel observations and states, facilitating the evaluation of representation learning methods for pretraining. Alongside the dataset, the paper proposes three benchmarks to assess these methods, revealing insights into policy learning under different conditions.

---

> ### Author Rebuttal · Authors · 2024-08-15
>
> Thank you for your feedback on our work. We are heartened that you consider that our work *“significantly advances the field of offline RL”*.
>
> > Lack of the complex control tasks limit the quality of this work. Maybe you can try some robotics task?
>
> As our goal is to create a benchmark for representation learning for control, we have built on the DM control suite which is commonly used to develop novel RL algorithms. Despite its simplicity in some respects,  DMC-VB is designed to include a diversity of tasks each with various difficulties up to *“tasks where state-of-the-art algorithms struggle”* (line 49). In particular, as seen in Figure 3, both the DMC-VB tasks of (a) humanoid locomotion with visual distractors and (b) ant maze navigation with multiple obstacles remain challenging for state-of-the-art visual control algorithms.
>
> As we acknowledge in the limitations (lines 305-309), although DMC-VB includes a diverse set of locomotion and navigation tasks, it could be extended to a broader range of environments. In particular, we are interested in onboarding robotics tasks in the future.

---

### Official Review · Reviewer_soTG · 2024-07-27
**Review of DMC-VB**

**Rating:** 7
**Confidence:** 3
**Correctness:** Yes, the claims in the submission are…

**Review:**

The paper is well written with an extensive study of the dataset constructed, considering a wide range of approaches for both visual representation learning + algorithms for policy learning (imitation learning/RL). The motivation of the work is quite clear and providing a benchmark to study this at a smaller scale is useful to study the efficacy of different representation learning algorithms. The evaluation of the benchmark in 3 regimes makes quite a bit of sense to study different aspects of the representation learned.

**Strengths:**

- Clear pipeline for how the dataset is constructed for each of the 3 regimes found in the benchmark
- Extensive evaluation over pre-trained visual representations and policy learning approaches.
- Provide an approach to study the representations learned in pre-training
- Ablation over different dataset compositions (such as mixed data, stochastic goals) to study the performance of different algorithms in different settings.

**Additional Feedback:**

Great paper. Would be amazing to hear more about the concerns I had highlighted in the improvement section. Thank you!

**Clarity:**

The paper is well written with clean figures/captions and a well-motivated story.

**Documentation:**

Yes, there is sufficient detail in how the data is collected.

**Ethics:**

No ethical concerns found.

**Limitations:**

Yes, limitations are clearly addressed in the work.

**Opportunities For Improvement:**

- It is disapointing to see that visual representation learning seems be quite similar to or slightly hurt performance in comparison to training end-to-end from scratch in Figure 3 and 5. A clarifying question is whether visual distractors are found in the datasets that were used to train the pre-trained representations that were learnt. Given that the observation/state reconstruction being low, it is hard to reconcile why this doesn't lead to better downstream policy performance. Are the pre-trained features here potentially frozen? If so, are there gains when fine-tuning these features?
- Would be useful to consider alternative offline RL algorithms (e.g IQL/CQL) or offline-to-online RL algorithms to see if we can improve performance with online samples.
- For the antmaze domains, could be useful to consider the coverage of the mixed vs expert datasets. A potential reason that suboptimal data could be helpful for the agent to perform better, is the higher coverage over the state space allowing the agent to recover from states that is OOD from the expert data state distribution.

**Relation To Prior Work:**

The work clearly distinguishes their approach from prior work, highlighting 6 categories in which they differ.

**Summary And Contributions:**

This work constructs a benchmark to study offline RL and imitation learning algorithms in the visual representations that they learn. The benchmark is constructed utilizing the DeepMind Control Suite to study 6 key properties not all found in prior work: (1) Task diversity, (2) Distractor diversity, (3) Different policies for demonstration collection (4) Utilizing Images + State observations, (5) Larger than previous datasets, and (6) has tasks where the goal cannot be determined from visual observations. The paper discusses several key details regarding the dataset, such as the dataset construction, performs extensive baseline evaluation, and constructs 3 dataset subsets to study different properties of visual representation learning such as robustness to distractors, datasets of mixed quality, and stochastic hidden goals. Additionally, an architecture ablation is done to study the efficacy of pre-trained visual representations both on downstream performance as well as observation/state reconstruction.

---

> ### Author Rebuttal · Authors · 2024-08-15
>
> Thank you for your positive feedback on our work.
>
> > A clarifying question is whether visual distractors are found in the datasets that were used to train the pre-trained representations that were learnt.
>
> In lines 120-126, we detail the types of visual distractors in our datasets. In particular, visual distractors are present in all datasets apart from those labeled “None" for the visual distractor type (see Figure 3 top). Note that *“when visual distractors are present, the evaluation environment contains unseen visual distractors from the training (dataset) distractor distribution”* (lines 223-224).
>
> > Given that the observation/state reconstruction being low, it is hard to reconcile why this doesn't lead to better downstream policy performance. Are the pre-trained features here potentially frozen? If so, are there gains when fine-tuning these features?
>
> Yes, we freeze the visual encoder when we learn the policy, as outlined in lines 164-165 or 208-210. We do not fine-tune the encoder during policy learning as this is more common in the literature—e.g, [1] [2]—and is a cleaner way to evaluate the pre-trained representations. We expect that fine-tuning the encoder would lead to similar findings and potentially reduce the disparity between pre-training methods.
>
> Regarding reconstruction error and performance, it is important to distinguish between state and observation (pixel) reconstruction error. Our findings reveal that representations that can attain low **state** reconstruction error lead to better downstream policy performance, however representations that attain low **observation** reconstruction error do not necessarily. Accurate state reconstruction correlates well with performance as the state contains the minimal features necessary for control. On the other hand, representations that can accurately reconstruct the observation will not be robust to visual distractors and therefore may result in worse downstream policies. This point is conveyed in Figure 4 which shows that the representations learned by ID or NULL+BC (which does not use pretraining and directly learns a policy with BC), achieve the lowest state reconstruction errors and lead to the best performing policies.
>
> We have added the following text to Section 5.1 within “Inspecting the Visual Representation” to make this point clearer in the revised paper: *“Fig.4 shows that representations that achieve low state reconstruction error capture minimal control-relevant features and lead to better downstream policy performance (ID, NULL+BC). Meanwhile, representations that attain low observation reconstruction error are not robust to visual distractors and result in worse downstream policies (LFD, AE, DINO).”*
>
> > Would be useful to consider alternative offline RL algorithms (e.g IQL/CQL) or offline-to-online RL algorithms to see if we can improve performance with online samples.
>
> We consider the offline RL method TD3-BC ([https://arxiv.org/pdf/2106.06860](https://arxiv.org/pdf/2106.06860)) in our benchmark evaluations. Results for TD3-BC are presented in Figures 3 and 8, as well as in Table 4, Appendix E.1. As TD3-BC was outperformed by BC, we did not include TD3-BC in all experiments, only using BC for clarity of presentation. TD3-BC is implemented in our open-source code [here](https://github.com/google-deepmind/dmc_vision_benchmark/blob/main/dmc_vision_benchmark/rep_learn/td3_bc.py).
>
> We thank you for your suggestion and we would like to add other offline RL algorithms, in particular CQL, to our GitHub repository in the future. More generally, we hope that the research community will leverage our benchmarks to explore combinations of representation learning methods with other offline and online RL algorithms.
>
> > For the antmaze domains, could be useful to consider the coverage of the mixed vs expert datasets. A potential reason that suboptimal data could be helpful for the agent to perform better, is the higher coverage over the state space allowing the agent to recover from states that is OOD from the expert data state distribution.
>
> We agree and found in our experiments that data coverage and diversity is important for performance. For this reason, we did not define the expert behavioral policy as the last checkpoint and instead *“we use the 95% reward checkpoint as the expert policy because we find that behavior diversity diminishes with more training steps”* (line 133). Note that as a way to measure the coverage of each dataset, we plot the reward distribution of all tasks for each behavioral policy level in Figure 2.
>
> We have added the following sentence to Section 5.1 to further elaborate the point of dataset coverage: *“We observe that for many tasks, performance degrades more strongly with visual distractors for expert than for medium datasets; we expect this is due to the medium dataset having higher coverage (see Fig. 2), reducing the chances of failure by entering a highly out of distribution state.”* Thank you for raising this comment!
>
>
> **References**
>
> [1] Islam, Riashat, et al. "Agent-controller representations: Principled offline rl with rich exogenous information." arXiv preprint arXiv:2211.00164 (2022).
>
> [2] Brandfonbrener, David, Ofir Nachum, and Joan Bruna. "Inverse dynamics pretraining learns good representations for multitask imitation." Advances in Neural Information Processing Systems 36 (2024).

---

### Author Response · Authors · 2024-08-15

We thank the reviewers for their helpful comments. We are glad that all reviewers find the dataset and benchmarks to be valuable contributions to the offline RL community.

---

### Decision · Program_Chairs · 2024-09-26

**Decision:**

Accept (Poster)

**Comment:**

This work proposes a benchmark to study offline RL and imitation learning algorithms in visual representations. All reviewers consistently recommended accepting this work. AC agrees that this work is interesting and deserves to be published on the NeurIPS dataset track 2024. The reviewers did raise some valuable concerns that should be addressed in the final camera-ready version of the paper. The authors are encouraged to make the necessary changes in the final version.